# Peer review of "Supporting Machine Learning Model in the Treatment of Chronic Pain"

_biomedicines, 2023, doi:10.3390/biomedicines11071776_

Round 1
Reviewer 1 Report
Plants of the hemp genus (Cannabis L.) are very important a raw material, f.e. they were introduced to homeopathy in the year of 1841. In high doses it evokes euphoria and hallucinations. These properties are exactly why it was used as treatment for various illnesses. Among main pharmacologically active substances of hemp drugs belong cannabinoids. Different therapeutical and toxic activity of various types of natural substances and their mutual interactions in human organism lead to complicated endocannabinoid system. In the year of 1988 and subsequently in 1993 where in central nervous system discovered receptors with high affinity to phytocannabinodoids. These significant discoveries jump-started research into new human signaling system. Most scientific papers are focused on two main representatives of hemp metabolites, this are THC /(–)-Δ9-trans-(6aR, 10aR)-tetrahydro-kanabinol/ and CBD / (–)-trans-(3R, 4R)-kanabidiol/.
The aim of this manuscript is showed the machine learning techniques, which is demonstrated satisfactory performance on a wide range of tasks in biomedicine, including knowledge improvement in pharmacogenetics.
My notes and comments on the text are as follows:
1. The introduction is too extensive.
2. Figure 2. is confusing and insufficiently explained.
3. The conclusion is too broad and doesn’t use enough references. It’s necessary to state more important facts from the research conducted. Prospectively, it’s necessary to add information about the next research project and its aim.
In regard to my opinion the contents of the manuscript in line with policy of the journal, the text has been prepared according to the format and style of journal including the body of the manuscript, page size and referencing. The manuscript should be accepted in the form after its correction.
Minor editing of English language required
Author Response
We would like to thank the Editor and the Referees for their insightful comments and constructive feedback on our review paper “Supporting Machine Learning Model in the cannabis therapeutic process”. We have carefully addressed all the comments and revised our manuscript accordingly. The added/modified parts in the revised version of the manuscript are highlighted in yellow.

Reviewer 2 Report
The authors of the reviewed manuscript presented the results of their analyses, which, in my opinion, fit in very well with the modern approach to therapy within personalized medicine, using pharmacogenetics. The research undertaken by the authors has important implications for the implementation of correct and safe pharmacotherapy using medical cannabis. After carefully reviewing the manuscript, from my point of view, the authors should consider the following suggestions, which may increase the scientific value of the presented manuscript:
1. I noticed some discrepancies in the authors' affiliations. It is important to make the necessary corrections to accurately represent their affiliations and ensure proper recognition of their institutional connections.
2. The results presented in the manuscript do not cover a large group of patients despite being conducted over a seven-year period. In their earlier publication (https://doi.org/10.3390/genes13101832), authors elaborated more on the research topic they had undertaken. Therefore, I believe that the title should include a statement about the preliminary analysis of the supporting role of machine learning models in the cannabis therapeutic process.
3. While the presented data bring some elements of novelty, I believe the authors should place greater emphasis on the criteria that limit their results.
4. I noticed that there are instances where the capitalization of disease names is inconsistent, particularly in line 38 and 39. This inconsistency should be addressed for the sake of clarity and adherence to proper conventions.
5. The descriptions accompanying Figures 2, 4, and 5 could benefit from further clarification. Providing clearer descriptions would greatly enhance the interpretability of these figures.
6. The authors' presented discussion appears to be more akin to conclusions rather than a comprehensive discussion. In my opinion, it requires refinement and expansion to provide a more thorough analysis of the findings.
7. It would be beneficial to include more up-to-date references in the manuscript. This would strengthen the relevance of the work and demonstrate a comprehensive understanding of the current state of research in the field.
Author Response

(The authors gave the same response as above.)

Round 2
Reviewer 2 Report
After carefully reviewing the manuscript, I believe that the revisions made by the authors have significantly enhanced its scientific value, making it well-suited for publication. The authors have clearly put in a lot of effort to improve the clarity and coherence of their ideas, which has resulted in a more compelling and robust study. The manuscript is now presented in a clear and concise manner, which will undoubtedly benefit readers and fellow researchers in the field. Overall, I am confident that the manuscript is now of high quality and will make a valuable scientific contribution.
Author Response
We would like to thank the Reviewer 2 for the thoughtful comments and efforts towards improving our manuscript.